# The Treatment of Acute Diaphyseal Long-bones Fractures with Orthobiologics and Pharmacological Interventions for Bone Healing Enhancement: A Systematic Review of Clinical Evidence

**DOI:** 10.3390/bioengineering7010022

**Published:** 2020-02-24

**Authors:** Giuseppe Marongiu, Andrea Contini, Andrea Cozzi Lepri, Matthew Donadu, Marco Verona, Antonio Capone

**Affiliations:** 1Orthopaedic and Trauma Clinic, Department of Surgical Sciences, University of Cagliari, 09124 Cagliari, Italy; andre.contini89@gmail.com (A.C.); marco.verona@tiscali.it (M.V.); anto.capone@tiscali.it (A.C.); 2Orthopaedic Traumatologic Center, University of Florence, 50121 Florence, Italy; andreaco84a@gmail.com; 3Dipartimento di Chimica e Farmacia, University of Sassari, 07100 Sassari, Italy; mdonadu@uniss.it

**Keywords:** shaft fractures, long-bones, bone healing, diamond concept, scaffolds, cell therapies, growth factors, teriparatide

## Abstract

Background: The healing of long bones diaphyseal fractures can be often impaired and eventually end into delayed union and non-union. A number of therapeutic strategies have been proposed in combination with surgical treatment in order to enhance the healing process, such as scaffolds, growth factors, cell therapies and systemic pharmacological treatments. Our aim was to investigate the current evidence of bone healing enhancement of acute long bone diaphyseal fractures. Methods: A systematic review was conducted by using Pubmed/MEDLINE; Embase and Ovid databases. The combination of the search terms “long-bones; diaphyseal fracture; bone healing; growth factors; cell therapies; scaffolds; graft; bone substitutes; orthobiologics; teriparatide”. Results: The initial search resulted in 4156 articles of which 37 papers fulfilled the inclusion criteria and were the subject of this review. The studies included 1350 patients (837 males and 513 females) with a mean age of 65.3 years old. Conclusions: General lack of high-quality studies exists on the use of adjuvant strategies for bone healing enhancement in acute shaft fractures. Strong evidence supports the use of bone grafts, while only moderate evidence demineralized bone matrix and synthetic ceramics. Conflicting results partially supported the use of growth factors and cell therapies in acute fractures. Teriparatide showed promising results, particularly for atypical femoral fractures and periprosthetic femoral fractures.

## 1. Introduction

Long-bones diaphyseal fractures are characterized by morphology patterns, with displacement and comminution, that could cause low bony contact, bone loss and vascular supply disruption, and consequently an impaired bone healing process. However, other factors, such as soft tissue damages, open fractures and patients’ related factors could lead to an increased risk of delayed unions and non-unions, which occur in up to 10% of all diaphyseal fractures [1,2]. Fractures at high risk of impaired bone healing should be identified according to fracture characteristics and patients’ related factors, in order to choose the best-suited treatment [3]. In the last decade, the so-called “Diamond Concept” proposed a new paradigm for complex fractures and impaired union management in which the main elements—mechanical environment, scaffolds, growth factors and cell therapies—could be applied together to enhance bone healing [4]. The primary goal of the treatment is to provide mechanical stability at the fracture site preserving the local vascular supply, therefore, diaphyseal fractures of long bones are mostly surgically treated. Then, additional local biological enhancement could be obtained by the addition of scaffolds, growth factors and cell therapies. These local therapies are part of the orthobiologics group of biological materials and substrates which have arisen in orthopaedics surgery to promote tendon, ligament, muscle and eventually in bone healing [5] (Table 1).

The evidence available showed convincing results supporting the use of “polytherapy”, although the majority of the studies exploited all the features of the diamond concept only for high-risk patients, delayed union and non-unions, which may have biased the results [6,7].

Pharmacological interventions that could modulate the healing process positively have been recently merged to the treatment algorithm, where systemic anti-osteoporosis drugs are considered the additional contributing factor [8]. The role of systemic drug therapies in long bones fracture healing remains limited, although the use several anabolic agents has reported positive results in selected shaft fractures types, such as atypical femoral fractures, periprosthetic femoral fractures and recalcitrant non-unions [9,10,11].

However, lack of evidence exists about the effectiveness of these treatments in acute fractures. Therefore, the purpose of this systematic review is to synthesize the available data about bone healing enhancement for acute diaphyseal fractures with use of the “diamond concept” approach and pharmacological interventions.

## 2. Materials and Methods 

A literature search related to bone healing enhancement of long bone fractures was performed. The main criteria for selection were English language articles focused on the role of clinical interventions for fracture healing including: grafts, bone substitutes, cell therapies, growth factors. Exclusion criteria were: non diaphyseal fractures and non-clinical studies. Moreover, additional search included studies focused on systemic pharmacological therapies for bone healing enhancement. 

Searches were conducted from January 2019 and July 2019 using the following databases: Pubmed/MEDLINE, Embase and Google Scholar. The combination of the search terms “femur”, “tibia”, “humerus”, “long bones”, “diaphyseal fracture”, “bone healing”, “fracture healing”, “diamond concept”, “growth factors”, “cell therapies”, “scaffold”, “bone graft”, “bone substitutes”, “orthobiologics” was used to find the literature relevant to the topic under review. A second search combined the previous terms with “antiosteoporosis drugs”, “teriparatide”, “biphosphonates.”

Scientific meeting abstracts and other sources of evidence were not considered. Titles and abstracts were reviewed for relevance by two independent authors. Reference list of trials and review articles were used to search additional sources. Those considered consistent with the aim of this review were read in full text. (Figure 1). Considering the paucity of high-level studies on this subject, we even included case series and case reports in the review.

## 3. Results

The initial search resulted in 4156 articles, of which 37 papers fulfilled the inclusion criteria and were the subject of this review. This review comprised 12 randomized controlled trial (RCT) (Level II), four retrospective comparative study (Level III), 12 retrospective case series and eight case reports (Level IV). Moreover, one systematic review with one prospective non randomized non controlled study (Level III), three retrospective case series and seven case reports (Level IV) were included. The aforementioned studies included 1350 patients in total. The mean age of the patients included in this review was 65.3 years old (42–87), while more than 62% of them were males (837 males and 513 females). 

The papers included in the review are reported by category in Table 2.

### 3.1. Autologous Bone grafting (Autograft)

Bone grafting techniques have a major role in several orthopedic and trauma procedures in which bone augmentation and regeneration is needed [12]. Autologous bone graft (ABG) combines all properties required for a biological graft: osteogenic, osteoconductive and osteoinductive properties. The application of cortical and cancellous ABG have been widely studied in non-unions of long bones shaft fractures with consistent union rates ranging from 80% to 94% [6,13]. The cortical ABG from the fibula or iliac crest can be both vascularized and non-vascularized [14]. A vascularized bone graft is harvested from the donor site with its vascular bundle, which is anastomosed to the recipient blood circuit and is then stabilized with an internal fixation to the recipient bone. In acute fractures, cortical structural bone grafts can act as biological plates or intramedullary support in combination with other internal fixation devices, improving fracture stability. For open tibial shaft fractures, with severe bone loss, vascularized ABG represents the treatment of choice (Table 3). Two different trials [15,16], used iliac crest autograft for large defect open tibia fractures with lower bone union rates: 81% (9/11) and 70% (7/10), respectively. Zhen reported 100% of union after plating and vascularized fibular osteo-septocutaneous flap graft for complex tibial shaft complex fractures; however, 10 out 38 experimented donor-site morbidities [17]. Moreover, bone grafting represents an additional option in all the osteoporosis related and so-called fragility fractures [10]. Limitations of ABG extensive use are the limited availability, graft dimension and donor site morbidity (i.e., iliac crest or fibula) [18]. 

Moreover, cancellous autograft could be obtained from the medullary canal of long bones. The reamer/irrigator/aspirator (RIA) technique was described for long bones intramedullary nail fixation with the aim to reduce the incidence of fat embolism and thermal necrosis related to reaming and nailing of shaft fractures [19]. During the procedure, a great amount of cancellous bone particulate and marrow contents are produced and collected for bone grafting. The use of RIA for acute diaphyseal long bone fractures did not show favorable results when compared to standard reaming in the rate of healing or overall union rates [20,21]. Therefore, at the moment the main clinical applications are the treatment segmental bone defects [22], osteomyelitis and non-unions [23]. 

### 3.2. Allogenic Bone grafting (Allograft)

Allogenic bone grafts (Allograft) are available in different shapes and dimension as cortical, cancellous, osteochondral and whole-bone segments. The main disadvantage of allografts is the loss of osteogenic potential and diminished osteoconductive and osteoinductive characteristics due to the removal of bone cells during the graft preparation. In fact, in order to avoid a potential antigenic response and infectious agents transmission, allografts are processed in several forms such as mineralized or demineralized, fresh, fresh-frozen or freeze-dried [24]. Freeze-dried and fresh-frozen bone allografts induce better graft vascularization, integration and bone regeneration than fresh allograft. However, allografts take longer to achieve revascularization and osseointegration compared to autografts. Moreover, these processing procedures have high costs and also reduce the mechanical strength of the graft. Generally good results are reported for complex humeral and femoral shaft fractures (Table 3). In a 45 patients’ series, structural allograft associated to open reduction and internal fixation (ORIF) for complex humeral and femoral shaft fractures provided 93% and 88% union rates, respectively [25].

According to several studies cortical strut allografts represent a reliably solution in the treatment of periprosthetic femoral fractures (PFF) in patients with a hip arthroplasty [26,27,28]. Particularly, grafts have been applied in periprosthetic femoral fractures, classified as type B1 (fracture around a stable femoral stem) of the Unified Classification System (UCS), in type C (fracture below the stem) and in re-osteosynthesis of a previously failed fixation, with union rates ranging from 92% to 100% [26,29,30,31,32,33,34,35,36]. In a systematic review reporting the outcome of B1 fractures, no significant difference in healing rates between plate fixation alone and plate fixation with strut graft were found (91% vs. 92%, *p* < 0.05). Therefore, due to higher risk of infection, allograft structural graft should be applied with caution for the treatment of B1 periprosthetic factures [37] (Table 3).

### 3.3. Bone Substitutes

The demineralized bone matrix (DBM) is an allograft in which the inorganic mineral is removed, while the organic matrix, such as collagenous and non-collagenous peptides and growth factors, are preserved [38]. DBM, as a biomaterial, provide both osteoconductive and osteoinductive healing. The osteoconductivity properties is guaranteed by the three-dimensional structure of the DBM, which promote cells infiltration and bone formation. The osteoinductive feature is represented by the stimulation of mesenchymal cells (MSC) by residual bioactive molecules of the DBM. MSC eventually differentiate into osteoblasts and initiate the process of endochondral ossification at the fracture site [39]. DMB, due to its inferior structural mechanical integrity than ABG and strut-allograft is applied for filling voids of bone defects [40]. Particularly, results in long bones fracture surgery have shown good results (Table 3). Lindsey et al. reported 90% bone healing after DBM augmentation compared to 75% after iliac crest autograft in 20 consecutive long bone fractures (humerus, femur or tibia) underwent to open reduction and internal fixation [41]. In another report, Kulachote et al. showed shorter healing time when atypical sub-trochanteric femoral fractures were treated with DBM in addition to intramedullary nail (IMN) fixation. The nine sub-trochanteric fractures treated with DMB healed in a mean of 28.1 ± 14.4 weeks versus 57.9 ± 36.8 of the nine sub-trochanteric fractures of the non-DMB group (*p* = 0.04) [42]. However, studies supporting the use of DBM in long bone shaft fractures are represented by retrospective case-series and provided only poor clinical evidence [43]. 

Calcium sulphate, calcium phosphate (CaP) ceramics and injectable cements and bio-active glass, represent the more used synthetic bone substitutes. These composites share similar mechanical characteristics, aiming to imitate the osteoconductive function of autografts and allografts, and are mainly used to fill the void in large segmental defects [5,44]. 

Ayoud et al. retrospectively studied the results of β-tricalcium phosphate combined to DBM putty, as a primary hybrid grafting, in 62 long bone comminuted fractures (femoral, humeral and tibial) treated by plate fixation. The authors claimed a 100% healing rate without any implant failure, reporting delayed union only in case of critical-size defects [45]. Chapman et al. in 1997 reported the results of a RCT comparing a collagen-calcium phosphate ceramics graft material with iliac crest autografts for diaphyseal long bone fractures [46]. There were 202 diaphyseal fractures randomized in the two groups. At a 24-month follow up, the authors reported 98% of union rate in the ceramic grafts group and 99% in the autografts group, without statistically significant differences. Overall complications rates were similar; however, the rate of infection was higher in the autograft group than that in the collagen-ceramic graft group (13% vs. 4.9%). The application of bioactive glass showed promising results [47]. Sun et al. in an RCT, compared the use of reamed IMN alone versus reamed IMN combined with Bioglass 45S5 grafting at the fracture sites for the treatment of 78 tibial diaphyseal fractures. All the patient in the experimental group healed within 6 months, while 4/38 patients (10.5%) in the control group underwent delayed union [48]. Moreover, these bone substitutes can be applied as scaffolds if combined with osteoinductive molecules, such as BMPs and osteogenic substrates such as bone marrow aspirate and PRP [5].

### 3.4. Growth Factors and Peptides

Growth factors are a set of proteins that play an essential role in repair and regeneration processes, include Bone Morphogenetic Proteins (BMP), vascular endothelial growth factor (VEGF), fibroblast growth factors (FGF), insulin-like growth factor (IGF) and platelet-derived growth factor (PDGF). They have been proposed with regard to their stimulation activity of bone healing and used for clinical application. The group of Bone Morphogenetic Proteins (BMP), particularly BMP-2, BMP-4 and BMP-7, has found wide application in the orthopaedic field [49,50,51]. BMPs recover a main role in the stimulation of bone healing. They lead the mesenchymal cells to the differentiation into osteoblasts and stimulate chondrocytes during endochondral phase of bone healing. In experimental models of reparative osteogenesis, BMP-2 and BMP-7 enhanced the production of cartilage and woven bone within the callus, reducing the overall time needed to close the fracture gap (i.e., hard callus and osseous remodelling) [52,53]. Two RCTs reported that when BMP-2 and BMP-7 combined with internal fixation of tibial shaft fractures provide a better healing rate, and less time was needed to lead the fracture to a complete union [54,55]. Govender et al. in an RCT of 450 open tibia shaft fractures, showed that the application of 1.50 mg/mL recombinant (rh) BMP-2 was useful in order to reduce the healing time and the risk of non-union [56]. Other studies have shown controversial results. Aro et al. used an absorbable collagen sponge containing rhBMP-2 for open tibial fractures treated with reamed intramedullary nail fixation. They did not find significant benefits in terms of healing time compared to the control group [57]. Lyon et al. in patients with closed tibial fractures treated with IMN, found that time to union and full weight-bearing without pain were not significantly reduced by the use of 2.0 mg/mL of rhBMP-2/CPM compared with standard of care alone [58]. The application of BMPs to treat fracture non-unions, according to several authors appeared to be a favorable alternative to autologous bone grafting. Particularly, the injection or the use collagen sponge carriers of BMP-2 or BMP-7 in the non-union site resulted in similar healing rates (73–89%) as those achieved with autografts [59,60]. As reported by Klenke and Siebenrock, interestingly, RCTs investigating the application of BMPs and autograft together have not been published thus far [61]. After the initial enthusiasm, several concerns have been raised about the use of BMPs. The induction of heterotopic ossifications has been reported from 9–18% of tibia fractures treated with BMPs, even if conversely other studies did not show a direct correlation [57,58]. Moreover, other morbidities are associated with its use, particularly in off-label and high dosage use, (due to its massive leakage from the carrier sponge into the soft tissue and then the systemic diffusion into the vascular flow) including compartmental syndrome, the potential for carcinogenesis, renal and hepatic failure [62,63]. In conclusion, the actual evidence seems to limit the use of rhBMP-2 to the treatment of severe open tibia fractures injuries, and the use of rhBMP-7 for treating tibia shaft non-unions [62] (Table 4).

### 3.5. Cell Therapies

Cell therapies used in the setting of bone healing enhancement consists in autologous bone marrow aspirate concentrate (BMAC) and Platelet rich plasma (PRP).

The autologous bone marrow aspirate concentrate (BMAC) contains red bone marrow, hematopoietic stem cells and mesenchymal stem cells (MSCs). The MSCs have the potential capability to trigger bone repair trough cell proliferation and differentiation. However, the concentration of MSC in bone marrow is less than 1 on 100,000 cells or less than 600 progenitor cells/cm^3^ [65], therefore, the main objective of BMAC is to concentrate as much as possible the number of progenitor cells contained in the aspirate. Hernigou et al. found that bone marrow from Iliac crest aspiration, after concentration, contained more than 2500 progenitor cells/cm^3^ [65]. Data from animal studies that suggested an MSC concentration of 10^6^ cells per mL and other human studies settled the MSC concentration limit at 2 × 10^7^ cells per mL. Higher concentrations are at risk for rapid resorption from within the graft area [66]. 

There is no consensus about the indications of BMAC for the treatment of fresh fractures (Table 4). Some authors described the use of BMAC injection in the fracture site or BMAC-enriched allografts for complex shaft fractures as alternative to autografts [67,68]. Le Nail et al. in a case series of 43 open tibia diaphyseal fractures, reported that poorer results were obtained when BMAC injection was performed in an early period after the trauma [69]. On the other hand, the role of BMAC in delayed unions and non-unions of shaft fractures is well established. BMAC can be applied both alone or in combination with enriched scaffolds (autografts, allografts and DBM), PRPs and BMPs. Different authors have reported that BMAC injection has a success rate ranging from 75% to 90% in treating atrophic non-unions of the tibia and humerus [65,70,71]. Unfortunately, no studies are reported in the literature that compare the use of BMAC injection with other non-union treatment techniques, (i.e., dynamization after IMN, exchange nailing or compressive plating with autografts).

Platelet rich plasma (PRP) is an autologous blood concentrate suspension of platelets obtained via different centrifugation techniques. Its use in the treatment of ligamentous and musculoskeletal injuries have yielded enthusiastic reports in the last decades. The aim of the application of PRP in bone healing is the attempt to mimic and augment the biological function of the hematoma at the fracture site. Local injection of PRP deliver platelets containing and releasing cytokines, such as Platelet-Derived Growth Factor (PDGF), Transforming Growth Factor Beta-1 (TGF-b1), Epidermal Growth Factor (EGF), Vascular endothelial growth factor (VEGF), Fibroblast Growth Factor (FGF), and Insulin-like Growth Factor (IGF), which have been addressed to enhance the healing process of injured tissues [72]. 

Only a few trials reported the clinical application of PRP for acute fracture of long bones. Liebergall et al. enrolled 24 consecutive patients with fractures of the distal tibial diaphysis, in a randomized controlled trial [66]. In the study group, the patients received MSCs from iliac crest bone marrow and peripheral blood and PRP, which were combined with DBM and used as a scaffold. The composite was then injected under X-ray control into the fracture gap. The mean time to union was 1.5 months in the study group and 3 months in the control group. Moreover, in the control group, 3 out of 12 patients experienced delayed union. In this study the specific contribution of each component of the composite graft is hard to quantify. The reported mean number of platelets in the platelet-rich plasma was of 1.1 × 10^9^ per concentrate. However, despite the number of cells introduced into the fracture site with PRP, their actual role and needed number of cells to achieve a biological activity remain unknown. In this setting, PRP may have acted as a scaffold enabling MSCs to remain in the fracture site.

More recently, Singh et al. in a randomized controlled trial, studied the efficacy of PRP on 72 acute fractures of the femoral shaft, treated with locked IMN [73]. The authors did not find a significant difference between the PRP group and control group according to the mean time to union, healing rate and complications rate. More favorable results were reported for the treatment of delayed unions and non-unions of long bones. In a prospective study, PRP for the treatment of 94 shaft fracture non-unions (35 of the tibia, 30 of the femur, 11 of the humerus, four of the radius, 12 of the ulna, two involving both radius and ulna) resulted in an 87% rate of union at 4 months [74]. In a randomized controlled trial, PRP injections were compared with exchange nailing after IMN, for the treatment of 29 diaphyseal oligotrophic non-unions of tibia and femur, showing superior healing rate in the PRP group compared with exchange nailing group (93% vs. 80%) [75]. Despite early promising results, at the moment, the grades of recommendation for PRP use in bone healing indicates conflicting or poor-quality evidence, and therefore, further investigation is needed [76]. 

### 3.6. Systemic Pharmacological Therapy

The process of consolidation and bone remodeling of a fracture is strictly dependent by bone turnover and the calcium and phosphate metabolism. According to the interesting findings on fragility fractures [9,77,78], current research is based on the off-label use of two classes of anti-osteoporotic drugs, antiresorptive and anabolic agents, for complex diaphyseal fractures that require enhancement of bone repair, such as delayed unions and non-unions [79] (Table 5). The use of these drugs on long bone diaphyseal fractures has not been adequately addressed by high-level clinical studies. 

#### 3.6.1. Bisphosphonates

The use of bisphosphonates (BP) for stimulating fracture healing is controversial. As antiresorptive drugs, inhibit bone resorption by blocking the action of osteoclasts, and this may cause a delay in bone remodeling. Other off-label use of BP showed beneficial results for the treatment of bone marrow edema [80]. Animal studies showed larger callus and stronger mechanical strength of femoral shaft fractures after bisphosphonates administration in ovariectomized rats [81]. Post-fracture bisphosphonate therapy on osteoporotic women with distal radius and proximal humeral fractures lead to increased BMD at the fracture site compared to placebo [82,83]. Moreover, the use of bisphosphonate in osteoporotic showed improved fixation stability at the bone—implant interface [84]. However, weak evidence exists on the effects of bisphosphonates either to reduce the time to union or enhance an impaired healing process, and at the moment no clinical study reported results of its human use in long bones diaphyseal fractures [85]. 

#### 3.6.2. Denosumab

Denosumab (DMAB) is a monoclonal antibody against the receptor for nuclear factor-kappa B ligand (RANKL) and has a strong antiresorptive activity. The biggest evidence on the effect of DMAB for fracture healing originated from a subgroup analysis of the FREEDOM study, a double-blind, randomized controlled trial enrolling post-menopausal women [86]. Results around DMAB and bone healing enhancement in non-vertebral fractures were substantially neutral: the early administration of DMAB after the fracture did not negatively affect the bone healing process or led to other complications. Even for DMAB, clinical studies reporting results of its human use diaphyseal fractures are lacking.

#### 3.6.3. Strontium Ranelate

Strontium ranelate is an antiosteoporotic drug which acts as an anabolic agent through two main mechanisms: it promotes bone formation increasing the action of osteoblasts and, on the other hand, reduces bone resorption limiting the action of osteoclasts [87]. Results from experimental studies on ovariectomized rats suggested the tibia fractures treated with strontium ranelate heal faster, with abundant callus and with higher BMD. Regarding the clinical use of strontium ranelate for fracture healing, the evidence is limited to clinical cases that suggest a potential benefit in complicated long bones fractures, atypical fractures and periprosthetic femoral fractures [11,88,89,90]. 

#### 3.6.4. Parathormone Analogues

Currently, two parathormone (PTH) analogues, PTH 1–34 (or teriparatide) and PTH 1–84, are available for clinical treatment of osteoporosis [91]. The application of these osteoanabolic agents for bone healing purposes in shaft fractures of animal models have shown findings indicating the increase of callus volume, improved mineralization and mechanical strength [92,93]. A wide range of different dosages and duration of the treatment have been proposed for both animal and human models [94]. Although early successful clinical results were reported in two RCTs, which analyzed the effect of PTH (1–34) vs. placebo in the treatment of distal radius and pelvis fractures [95,96], evidence that PTH peptides clinically improve fracture healing of long bones is limited to case reports or small case series. 

Recently, research has focused on the use of teriparatide for the treatment of atypical femoral fractures (AFFs) [97]. AFFs are transverse sub-trochanteric and diaphyseal fractures of the femur (located between the lesser trochanter and the ishtmus) that occur after minimal trauma or in absence of trauma and are associated with long-term bisphosphonate or denosumab use. Several studies supported the use of PTH 1–34 in promoting bone union in AFFs [98,99,100]. Miyakoshi et al. [98] retrospectively reviewed 45 consecutive AFFs in 34 patients who received long-term BPs. Among the patients who underwent surgery, those in the teriparatide-group had lower mean time to bone union than the non-teriparatide group patients ((5.4 ± 1.5 months vs 8.6 ± 4.7 months, *p* < 0.005). Moreover, delayed union or non-union rates were significantly lower in the teriparatide group (*p* < 0.005). Both Watts et al. [99] and Chiang et al. [100] prospectively evaluated patients with AFFs and a history of previous BPs therapy. In their series, the patients who received 20 μg of daily teriparatide for 6 to 24 months had an improvement in bone turnover markers serum levels, but, on the other hand, there was no significant effect on fracture healing. More recently, Greenspan et al. reported the result of a pilot randomized clinical trial of 13 patients, to determine the correct timing of teriparatide administration after an AFF [101]. They found a trend in better healing rates in the group of patients who received immediate therapy with teriparatide compared to the group who received teriparatide therapy with a 6-month delay. However, there were no significant differences in complications and one implant failure was reported in the 6-month delayed therapy group. 

Shimada et al. [102] and Ito et al. [103] reported the successful treatment of three cases of ulnar atypical fracture, with internal fixation, teriparatide and low-intensity pulsed ultrasound.

Teriparatide therapy has been proposed for its possible benefits in the treatment of another type of shaft fractures: periprosthetic femoral fractures (PFFs). Although encouraging results of case series are reported, a lack of high level clinical studies still exists [104]. Lee et al. reported the results of a 19 patients case series, in which teriparatide was used for the conservative treatment of minimally displaced PFFs around the stem. Fracture healing was obtained in 16 patients out of 19 (84%) after two and six months (mean, 3.5 months) [105]. Miura et al. claimed the efficacy of teriparatide in two Scases of atypical periprosthetic femoral fractures [106]. Similarly, other anecdotical cases were reported [107,108,109,110,111,112]. Kim et al. studied the radiographic features of bone healing after teriparatide therapy for PFFs, and found an early abundant callus formation in the first stage of treatment [113]. Therefore, they suggest to start treatment at day 1 after the trauma, if a conservative treatment is preferred. Ippolito et al. [114], in case report, applied teriparatide therapy for the treatment of a diaphyseal humeral fracture around a reverse shoulder arthroplasty. The fracture was stabilized through a mono-axial external fixator, then the patient received teriparatide at a dosage of 20 μg a day, for four months. At six months from the beginning of the treatment, a complete healing of the fracture was observed radiologically and clinically.

## 4. Conclusions

Although the evidence available showed convincing results supporting the use of “polytherapy” with orthobiologics treatments, especially for delayed union and non-unions, a few high-level studies and other reports showed conflicting findings in favor of the application of the “diamond concept” for acute diaphyseal fractures. In the current systematic review of the literature, we focused on the effect on grafts, synthetic scaffolds, growth factors and cell therapies for fracture healing combined with common fixation techniques. Cortical autografts, both vascularized and non-vascularized, represent the ideal system of local biological enhancement, by their osteoinductive, osteoconductive and osteogenic properties and mechanical support function. However, due to low availability and possible donor site morbidity, the use of ABG is almost dedicated to open diaphyseal fracture with severe bone loss. Structural allografts share similar characteristics with ABG (without osteogenic power) and are particularly employed when additional mechanical stability is needed such as in comminuted shaft humerus and femoral fractures and periprosthetic femoral fractures. A few high-level studies support the use of demineralized bone matrix and phosphate ceramics for acute tibial fractures with or without segmental bone defect. Other synthetic bone substitutes are often combined with cell therapies and growth factors, as a scaffold. In the group of growth factors, only rhBMP—2 have strong evidence of efficacy on acute fractures but restricted to open tibia fractures. rhFGF showed good healing rates combined to IMN of tibial shaft fractures. Although increasing interest has been showed around BMAC and PRP techniques, according to the current evidence, their application should be limited to the treatment of long bones non-unions. The role of systemic anti-osteoporosis drug therapies in acute long bones fracture healing remains limited, although teriparatide use has yielded positive results in selected shaft fractures types, such as atypical femoral fractures, periprosthetic femoral fractures and atypical periprosthetic femoral fractures.

In conclusion, further investigations and high-level studies are needed to assess the effectiveness of each of the different interventions and which combination would provide the best results in terms of bone healing rate, time to union and complications rate. Future efforts should be focused to reliably predict when a fracture is at high-risk for impaired bone healing and for selecting patients in whom the efficacy of therapeutic interventions to enhance fracture healing is assessed.

## Figures and Tables

**Figure 1 bioengineering-07-00022-f001:**
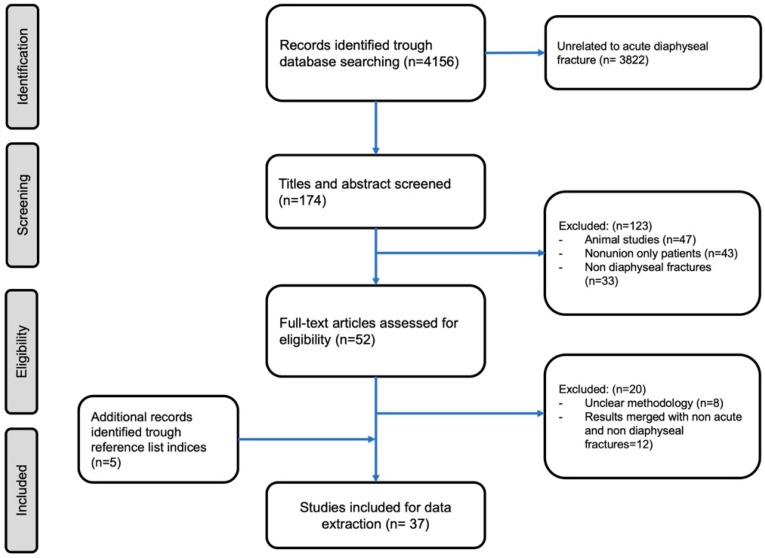
Our search strategy according to a PRISMA flowchart.

**Table 1 bioengineering-07-00022-t001:** Bone properties and efficacy of scaffold, growth factors and cell therapies in bone healing stimulation of diaphyseal fractures.

	Osteogenicity	Osteoconductivity	Osteoinductivity
**Autograft**	**++++**	**++++**	**++++**
**Allograft**	**-**	**+++**	**+**
**Demineralized Bone Matrix (DBM)**	**-**	**++**	**+**
**Calcium phosphate Hydroxyapatite**	**-**	**+**	**-**
**Bioactive glass**	**-**	**++**	**-**
**Bone Morphogenetic Proteins (BMPs)**	**-**	**-**	**+++**
**Platelet rich plasma (PRP)**	**+**	**-**	**++**
**Bone marrow aspirate concentrate (BMAC)**	**+++**	**-**	**++**

**Table 2 bioengineering-07-00022-t002:** Bone healing enhancement of acute diaphyseal fractures: literature summary by topic.

Topic	n. of Patients Included
**Bone graft** (15–17,25,26,29–33)	205
**Bone substitutes** (41,42,45,46,48)	331
**Growth factors** (53,55–58)	615
**Cell therapies** (66,69,73)	69
**Pharmacological therapy** (97)**(98–105,107–114)	110

** Systematic review including case series and case reports.

**Table 3 bioengineering-07-00022-t003:** Summary of clinical evidence of scaffolds effect on bone repair of acute long bones shaft fractures.

Type	Subtype	Clinical Evidence	Quality of Evidence
**Autologous bone graft (ABG)**	Cortical graft	Both vascularized and non-vascularized cortical graft showed effectiveness in tibia and humeral shaft fractures when associated to Open Reduction and Internal Fixation (ORIF). Vascularized ABG represent the treatment of choice for open tibia fractures with bone defect. High incidence of donor-site morbidities (i.e., iliac crest)	Strong
	RIA	Evidence suggest no effect in improvement of healing in acute fracture.	Moderate
**Allogenic bone graft (Allograft)**	Cortical graft	Cortical allograft showed effectiveness in complex femoral and humeral shaft fractures when associated to ORIF. Represent a reliably solution in the treatment of periprosthetic femoral fractures. High rates of infection	Moderate
**Bone substitutes**	Demineralized bone matrix (DBM)	Case series support the use in acute fracture of humerus, tibia and femur and atypical femoral fractures	Weak
	Calcium phosphate, β-tricalcium phosphate	Unclear benefit on fracture healing. Useful as scaffolds combined to DBM, growth factors and cell therapies	Weak
	Calcium phosphate ceramics	1 RCT showed that ceramics provide union rates similar to cortical autograft in acute long bones fractures. Lower rates of infection compared to autografts. In 1 RCT, Bioglass showed better healing rates for high-energy tibial shaft fractures, compared to the control group.	Moderate

In this table the level of clinical evidence is classified as “weak” when supported by level 3–4 studies. “Moderate” is supported by multiple level 2 studies or conflicting level 1 data. “Strong” evidence was determined by the analysis of multiple level 1 studies with consistent results. RIA: Reaming/Irrigation/Aspiration.

**Table 4 bioengineering-07-00022-t004:** Summary of clinical evidence of growth factors and cell therapies effect on bone repair of acute long bones shaft fractures.

Type	Subtype	Clinical Evidence	Quality of Evidence
**Growth factors**	BMP-2, BMP-7	rhBMP-2 for treating open tibia fractures, (^a^ Gustilo type 3) The rhBMP-7 is limited to treating tibia shaft non-unions. Potential local (heterotopic ossification) and systemic complications (carcinogenesis, renal and hepatic failure)	Strong
	rhFGF	Case series suggested benefit in tibial shaft fractures.	Weak
**Cell therapies**	Autologous bone marrow aspirate concentrate (BMAC)	Case series reported unclear benefit in bone healing of acute open tibia fracture.	Weak
	Platelet rich plasma (PRP)	Conflicting evidence. 1 RCT showed shorter healing time in tibial fractures + ORIF. 1 RCT showed no difference in femoral subtrochanteric fractures + IMN	Moderate

In this table the level of clinical evidence is classified as “weak” when supported by level 3–4 studies. “Moderate” is supported by multiple level 2 studies or conflicting level 1 data. “Strong” evidence was determined by the analysis of multiple level 1 studies with consistent results. ^a^ Gustilo type 3 fractures are open segmental fractures with open fracture with extensive soft tissue damage [64].

**Table 5 bioengineering-07-00022-t005:** Summary of clinical evidence of systemic drugs’ effect on bone healing of long bones shaft fractures.

Agent	Class	Clinical Evidence	Quality of Evidence
**Calcium/vitamin D**	Supplemental	Potential activity of increasing bone mineral density at fracture site	Weak
**Bisphosphonates**	Antiresorptive	Unclear evidence suggests inhibition of healing. Unclear if there are benefits in augmenting healing. Improve BMD at fracture site when administrated 2 weeks after fracture. Improve fixation at bone implant interface	Weak
**Denosumab**	Antiresorptive	Does not affect negatively bone healing Not clear if enhance bone healing	Weak
**Strontium ranelate**	Anabolic	Case reports shown favor on fracture healing. Augment callus resistance and volume. Reports showed enhancement in union after delayed union or non-union.	Weak
**Teriparatide**	Anabolic	Reduction of healing time in long bone fractures and improved implant stabilization. Moderate evidence for enhanced bone healing of delayed unions, non-unions, atypical femoral fractures and periprosthetic femoral fractures	Moderate

In this table the level of clinical evidence is classified as “weak” when supported by level 3–4 studies. “Moderate” is supported by multiple level 2 studies or conflicting level 1 data. “Strong” evidence was determined by the analysis of multiple level 1 studies with consistent results.

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
