# Peer review of "The Treatment of Acute Diaphyseal Long-bones Fractures with Orthobiologics and Pharmacological Interventions for Bone Healing Enhancement: A Systematic Review of Clinical Evidence"

_bioengineering, 2020, doi:10.3390/bioengineering7010022_

Round 1

Reviewer 1 Report

The following aspects of the paper should be improve before it is suitable for publication:

Table 1. Line 52. The expression "Mechanism of action" must be replaced by “Bone properties”. The authors show in the table the osteogenicity, osteoconductivity and osteinductive properties of different treatments, not the mechanisms of action of these treatments. I would suggest remove the final column of growth factors. I do not quite understand what the authors want to indicate with that column there. Growth factors are a type of treatment and are already mentioned in the first column, as BMPs and PRP.

Line 45. "Diamond concept", the authors should include a brief clarification on this concept.

Line 65. "Pentagon concept", the authors should include a brief clarification on this concept.

Line 94. In Table 2, and also in the text (Lines 162, 168, 184 and 187), appear abbreviations such as ORIF and RCT, which have not been described, so the reader does not know what they refer to.

Lines 100-101. The authors should include a brief explanation explaining what are vascularized and non-vascularized grafts and how they are obtained.

Line 130. The authors refer to specific types of periprosthetic fractures, but it is not indicated what this type of fractures consists of. The abbreviation UCS (Unified Clssification System) is not clarified. I would suggest the inclusion of a small picture with this classification, briefly indicating the types of periprosthetic fracture (A, B1, B2, B3 and C).

Líne 150, 168, 192, 228, 246, 252 and 358. The abbreviation IMN has not been described, so the reader does not know what they refer to.

Lines 202 to 203 "moreover ... syndrome (78,79)" in this paragraph, the authors mention the potential undesirable effects of using BMPs off-label and at high doses, such as carcinogenesis, and renal and hepatic failures. Perhaps at this point, the authors should mention that the two forms of BMP, -2 and -7 approved for human use, are formulated in collagen sponges and that the protein is released immediately with the subsequent systemic exposure, which could be the cause of the harmful side effects.

Line 175. I do not understand what the authors mean by the word “Synthetic”. In my opinion, the word “synthetic” refers to a molecule obtained through a synthesis process, not a molecule from natural origin such as growth factors. I suggest remove this word and start the paragraph like this. "Growth factors are a set of proteins that play an essential role in repair and regeneration processes, include ..."

Line 206. Table 4 refers to a type of tibia open fracture (Gustilo type 3). A brief explanation about this type of fracture should be included at the end of this table.

Line 269. The word “reabsorption” should be replaced by the word “resorption”.

Line 349. The word “osteogenetic” should be replace by the word “osteogenic”.

Author Response

Response to reviewer #1

The review paper presented by Marongiu et al "The treatment of acute diaphyseal long-bones fractures with orthobiologics and pharmacological interventions for bone healing enhancement: a systematic review of clinical evidence" presents a review of the different treatments available today, such as scaffolds, growth factors, cell therapies and systemic pharmacological treatments for bone healing enhancement of acute long bone diaphyseal fractures. The review work is exhaustive, with a significant number of clinical studies, randomized controlled trial, retrospective comparative studies, retrospective case series, case reports, etc.

The writing and wording is correct and fluid, which greatly facilitates the reading. Despite being an exhaustive review, it is not excessively long, which allows a quick reading and gives an integrated idea of the current perspective of the treatment of this type of fractures. In general, I have no important objections, however, it would be convenient that the authors improve some aspects in the paper before it is suitable for publication.

  • I really thank you for these encouraging words. I would try to address all of your concerns.

The following aspects of the paper should be improved before it is suitable for publication:

Table 1. Line 52. The expression "Mechanism of action" must be replaced by “Bone properties”. The authors show in the table the osteogenicity, osteoconductivity and osteoinductive properties of different forms of treatment, not the mechanisms of action of these treatments. I would suggest removing the final column of growth factors. I don't quite understand what the authors want to indicate with that column there. Growth factors are a type of treatment and are already mentioned in the first column, as BMPs and PRP.

  • We provided suggested changes in line 54 and the following table

Line 45. "Diamond concept", the authors should include a brief clarification on this concept.

  • We changed a few words in the text in order to better explain the theory as it follows

Line 44 – 53: “In the last decade the so-called “Diamond Concept”, proposed a new paradigm for complex fractures and impaired union management in which the main elements - mechanical environment, scaffolds, growth factors and cell therapies - could be applied together to enhance bone healing. The primary goal of the treatment is to provide mechanical stability at the fracture site preserving the local vascular supply, therefore diaphyseal fractures of long bones are mostly surgically treated. Then, additional local biological enhancement could obtained by the addition of scaffolds, growth factors, and cell therapies. These local therapies are part of the orthobiologics group of biological materials and substrates which have arisen in orthopaedics surgery to promote tendon, ligament, muscle, and eventually in bone healing.”

Line 65. "Pentagon concept", the authors should include a brief clarification on this concept.

  • I believe that the more exhaustive explanation of the diamond concept allows a better understanding of the role of the pharmacological interventions. However, since in the literature, the pentagon and the diamond concept are often used the explain the same theory we decided to do not mention in this review.

Therefore only we changed the text as it follows.

Line 60 – 63: “Pharmacological interventions which could modulate positively the healing process, have been recently merged to the treatment algorithm, where systemic anti-osteoporosis drugs are considered the additional contributing factor(7). The role of systemic drug therapies in long bones fracture healing remains limited, although the use of several anabolic agents has reported positive results in selected shaft fractures types, such as atypical femoral fractures, periprosthetic femoral fractures and recalcitrant non-unions (8–10).

Line 94. In Table 2, and also in the text (Lines 162, 168, 184 and 187), appear abbreviations such as ORIF and RCT, which have not been described, so the reader does not know what they refer to.

We’ve provide the needed clarifications.

  • At line 92 RCT is described in the text : “This review comprised 12 randomized controlled trial (RCT) (Level II), 4 retrospective comparative study (Level III), 12 retrospective case series and 8 case reports (Level IV)”

  • At line 129 ORIF is disclosed in the text for the first time in the phrase “In a 45 patients’ series, structural allograft associated to open reduction and internal fixation (ORIF) for complex humeral and femoral shaft fractures provided 93% and 88% union rates, respectively (14)”

Lines 100-101. The authors should include a brief explanation explaining what are vascularized and non-vascularized grafts and how they are obtained.

We agree with your point. Here you can find the change made.

  • At line 104 – 107: The cortical ABG from the fibula or iliac crest, can be both vascularized and non-vascularized. A vascularized bone graft is harvested from the donor site with its vascular bundle which is anastomosed to the recipient blood circuit and then is stabilized with internal fixation to the recipient's bone.

Line 130. The authors refer to specific types of periprosthetic fractures, but it is not indicated what this type of fractures consists of. The abbreviation UCS (Unified Classification System) is not clarified. I would suggest the inclusion of a small picture with this classification, briefly indicating the types of periprosthetic fracture (A, B1, B2, B3 and C).

Thank you for your analysis. We have provided a more clear explanation and we have corrected an error in the data reported. However, we believe that the inclusion of a figure with the UCS classification could excessively highlight this sub-topic over the others. Therefore, we would prefer not adding the figure.

  • At line 135 - 140: According to several studies cortical strut allografts represent a reliably solution in the treatment of periprosthetic femoral fractures (PFF) in patients with a hip arthroplasty (15,60,61). Particularly, grafts have been applied in periprosthetic femoral fractures, classified as type B1 (fracture around a stable femoral stem) of the Unified Classification System (UCS), in type C (fracture below the stem) and in re-osteosynthesis of a previously failed fixation, with union rates ranging from 92% to 100% (15–20,62).

Líne 150, 168, 192, 228, 246, 252 and 358. The abbreviation IMN has not been described, so the reader does not know what they refer to.

We provided a description of IMN.

  • At line 158 “In another report, Kulachote et al. showed shorter healing time when atypical sub-trochanteric femoral fractures were treated with DBM in addition to intramedullary nail (IMN) fixation”.

Lines 202 to 203 "moreover ... syndrome (78,79)" in this paragraph, the authors mention the potential undesirable effects of using BMPs off-label and at high doses, such as carcinogenesis, and renal and hepatic failures. Perhaps at this point, the authors should mention that the two forms of BMP, -2 and -7 approved for human use, are formulated in collagen sponges and that the protein is released immediately with the subsequent systemic exposure, which could be the cause of the harmful side effects.

We tried to address the issue in this phrase:

  • At line 210 – 213: “Moreover, other morbidities are associated with its use, particularly in off-label and high dosage use (due to its massive leakage from the carrier sponge into the soft tissue and then the systemic diffusion into the vascular flow) including compartmental syndrome, the potential for carcinogenesis, renal and hepatic failure”

Line 175. I do not understand what the authors mean by the word “Synthetic”. In my opinion, the word “synthetic” refers to a molecule obtained through a synthesis process, not a molecule from the natural origin such as growth factors. I suggest remove this word and start the paragraph like this. "Growth factors are a set of proteins that play an essential role in repair and regeneration processes, include ..."

We agreed to your suggestion. We removed it and changed the start of the paragraph. (line 183 – 186)

Line 206. Table 4 refers to a type of tibia open fracture (Gustilo type 3). A brief explanation about this type of fracture should be included at the end of this table.

Agreeing with your advice we added the description of Gustilo type 3 fractures and the related reference.

Line 269. The word “reabsorption” should be replaced by the word “resorption”.

We’ve changed the reabsorption in resorption

Line 349. The word “osteogenetic” should be replaced by the word “osteogenic”.

We’ve changed the osteogenetic in osteogenic

Reviewer 2 Report

In the Allograft section, it might be worth pointing out that the reason for removal of cells (and osteogenic potential) is because the allograft is not usually tissue matched and so any remaining cells mount an acute inflammatory/rejection response - and so have no osteogenic potential. This inflammatory response can be attenuated by freezing or removed by preparation to remove the cells such as washing or freeze-drying (or correctly tissue typing).

In the growth factor section, it would be worth discussing sham surgery, as some authors suspect that the act of needling a fracture has an effect, as well as actually injecting something. Many studies fail to have a control arm. This again may be relevant in the cell therapies section. Hernigou shows a cell concentration benefit however, which could be discussed versus PRP, which is rather more poorly quantified at the cell concentration level.

I'm not sure I get the point of excluding thousands of papers that done don't deal with just isolated acute diaphyseal fractures - but then to allow PPF's in the Teriparatide discussion. A diaphyseal fracture with or without an implant is often quite different. The presence of an implant (often uncemented) may confound this comparison (Diaphyseal fractures with or without an implant) as we know the long term outcome of joint replacements is slightly more favourable in patients on systemic therapy such as bisphosphonates (C Cooper et al). 

Author Response

Response to Reviewer #2

In the Allograft section, it might be worth pointing out that the reason for removal of cells (and osteogenic potential) is because the allograft is not usually tissue matched and so any remaining cells mount an acute inflammatory/rejection response - and so have no osteogenic potential. This inflammatory response can be attenuated by freezing or removed by preparation to remove the cells such as washing or freeze-drying (or correctly tissue typing).

  • Thank you for the interesting point, we added a brief explanation of the difference between Autograft and allograft, as you suggested.
  • At line 128 - 135: The main disadvantage of allografts is the loss of osteogenic potential and diminished osteoconductive and osteoinductive characteristics due to the removal of bone cells during the graft preparation. In fact, in order to avoid potential antigenic response and infectious agents transmission, allografts are processed in several forms such as mineralized or demineralized, fresh, fresh-frozen, or freeze-dried(61). Freeze-dried and fresh-frozen bone allografts induce better graft vascularization, integration, and bone regeneration than the fresh allograft. However, allografts take longer to achieve revascularization and osseointegration compared to autografts.Moreover, these processing procedures have high costs and also reduce the mechanical strength of the graft.”

In the growth factor section, it would be worth discussing sham surgery, as some authors suspect that the act of needling a fracture has an effect, as well as actually injecting something. Many studies fail to have a control arm. This again may be relevant in the cell therapies section. Hernigou shows a cell concentration benefit, however, which could be discussed versus PRP, which is rather more poorly quantified at the cell concentration level.

  • We agree that this is good speculation, particularly for nonunions, in which drilling and needling could revitalize the fracture site. However, in our opinion, this analysis could be poorly sustained regarding fresh fractures in which the fracture site is actually well vascularized. On the other hand we agree with you that despite the large number of cells introduced into the fracture bed with PRP, their actual role and needed number of cells to achieve a  biological activity remain unknown. We added in the text in cell therapies section, a dissertation about this topic.
  • Line 238 – 238: Data from animal studies that suggested an MSC concentration of 1 × 106 cells per ml and other human studies settled the MSC concentration limit at 2.0 × 107 cells per ml. Higher concentration is at risk to avoid overpopulation and rapid resorption from within the graft area(33).
  • Line 266 – 270: The reported mean number of platelets in the platelet-rich plasma was of 1.10 × 109 per concentrate. However, despite the number of cells introduced into the fracture site with PRP, their actual role and needed number of cells to achieve a biological activity remain unknown. In this setting, PRP may have acted as a scaffold enabling MSCs to remain in the fracture site.

I'm not sure I get the point of excluding thousands of papers that do don't deal with just isolated acute diaphyseal fractures - but then to allow PPF's in the Teriparatide discussion. A diaphyseal fracture with or without an implant is often quite different. The presence of an implant (often uncemented) may confound this comparison (Diaphyseal fractures with or without an implant) as we know the long term outcome of joint replacements is slightly more favourable in patients on systemic therapy such as bisphosphonates (C Cooper et al). 

  • We thank you for this comment. Our aim, including PFFs in the review, was to provide an answer to all the “acute” fractures that a trauma surgeon could have to face in his daily practice, due to the growing incidence of these fractures. We are aware that PFFs has a different biological underlying environment, however, we believe that PFFs which need fixation (type B1 and C) have biology more similar to a diaphyseal fresh fracture than, for example, to a delayed or a nonunion fracture. I hope that you can agree with our consideration.